# Genome-wide analysis of DNA G-quadruplex motifs across 37 species provides insights into G4 evolution

Feng Wu[1,2,7], Kangkang Niu[1,2,7], Yong Cui[1,2,7], Cencen Li[3], Mo Lyu[1,2], Yandong Ren[4], Yanfei Chen[1,2], Huimin Deng[1,2], Lihua Huang[1,2], Sichun Zheng[1,2], Lin Liu[1,2], Jian Wang[5], Qisheng Song [6✉], Hui Xiang [1,2✉] & Qili Feng [1,2✉]

G-quadruplex (G4) structures have been predicted in the genomes of many organisms and proven to play regulatory roles in diverse cellular activities. However, there is little information on the evolutionary history and distribution characteristics of G4s. Here, whole-genome characteristics of potential G4s were studied in 37 evolutionarily representative species. During evolution, the number, length, and density of G4s generally increased. Immunofluorescence in seven species confirmed G4s' presence and evolutionary pattern. G4s tended to cluster in chromosomes and were enriched in genetic regions. Short-loop G4s were conserved in most species, while loop-length diversity also existed, especially in mammals. The proportion of G4-bearing genes and orthologue genes, which appeared to be increasingly enriched in transcription factors, gradually increased. The antagonistic relationship between G4s and DNA methylation sites was detected. These findings imply that organisms may have evolutionarily developed G4 into a novel reversible and elaborate transcriptional regulatory mechanism benefiting multiple physiological activities of higher organisms.

[1] Guangdong Provincial Key Laboratory of Insect Developmental Biology and Applied Technology, Institute of Insect Science and Technology, School of Life Sciences, South China Normal University, Guangzhou 510631, China. [2] Guangzhou Key Laboratory of Insect Development Regulation and Application Research, Institute of Insect Science and Technology, School of Life Sciences, South China Normal University, Guangzhou 510631, China. [3] College of Life Sciences, Xinyang Normal University, Xinyang 464000, China. [4] Center for Ecological and Environmental Sciences, Northwestern Polytechnical University, Xi'an 710129, China. [5] Department of Entomology, University of Maryland, College Park, MD 20742, USA. [6] Division of Plant Sciences, University of Missouri, Columbia, MO 65211, USA. [7] These authors contributed equally: Feng Wu, Kangkang Niu, Yong Cui. ✉email: songq@missouri.edu; xiang_shine@foxmail.com; qlfeng@scnu.edu.cn

n addition to the classic DNA double-helix structure, tetraplex DNA secondary structures have been reported in cells[1]. G-quadruplex (G4) structures are DNA tetraplexes that typically form in guanine-rich regions of genomes. Four guanine bases associate with each other through Hoogsteen hydrogen bonds to form a guanine tetrad plane (G-quartet), and then two or more G-quartet planes stack on top of each other to form a G4 structure[2]. The complementary sequence of a G4 structure is a C-rich region, which correspondingly may form another secondary structure, called an i-motif[3]. DNA secondary structures were not a focus of research until they were found to be induced by cations in guanine-rich telomere sequences and affect the extension of telomeres[4,5]. Since then, this unique DNA structure has gathered substantial interest. Currently, it is well documented that quadruplex DNA, especially G4 structures, exists widely in the genomes of many organisms[6,7] and plays important roles in telomere protection, DNA replication, and gene transcription and translation[7–11]. More importantly, accumulating evidence indicates that G4 structures in the promoter regions of many genes are involved in the regulation of gene transcription[7,12–18].

Great efforts have been made to explore the genomic landscape and characteristics of G4 sequences, structures and distribution in plants[19,20], prokaryotes[21], viruses[22,23], and eukaryotes[24] by computational prediction, which indicate a widespread pattern in genomes of universal organisms and the functional significance of these structures in the promoter and 5′ regulatory regions of genes[25,26]. Two recent studies used high-throughput structural sequencing approaches to map genome-wide G4 structures in humans and other organisms[6,27]. Notably, in these studies, the vast majority (80–90%) of the G4 structures predicted by bioinformatics analysis were confirmed to exist in genomes by the G4 structural sequencing approach[6], suggesting the feasibility of prediction-based approaches.

One interesting issue is whether and how DNA secondary structures evolve as a regulatory mechanism of gene transcription and contribute to species evolution. In recent years, comparative analyses of the genomic landscape of G4 structures in multiple species have been reported[6,22,28,29]. For example, evolutionary conservation analyses of G4 structures in seven species of fungi revealed that G4 structures are relatively conserved throughout fungal evolution[28]. In addition, G4 structures were reported to be significantly enriched in transcriptional regulatory regions of warm-blooded animals by both prediction and experimental approaches[6,29]. How DNA secondary structures evolved during this large-scale evolutionary process from fungus to mammals is largely unknown.

Another interesting issue is that many G4-forming sequences within a genome harbour CpG sites, which are targets of DNA methylation, an important epigenetic system[30]. The relationship between CpG methylation and G4 motifs has recently been discussed in a few studies[31–33]. These two types of epigenetic regulation systems may interact with each other for fine gene regulation[33]. The abundance of available genome resources has enabled analyses of whether and how this relationship evolves at a large evolutionary scale.

In this study, a genome-wide G4 motif prediction was performed with 37 species at 14 representative evolutionary nodes that are critical in eukaryotic clade of the tree of life, covering a fairly large evolutionary scale ranging from single-celled fungi to higher mammals and humans; thus, the possible evolutionary patterns and functional implications of G4 motifs were comprehensively deciphered. The present study suggests that G4 structures have evolved with the evolutionary complexity of genomes and species, which enables higher organisms to achieve increasingly complex cellular, physiological, and behavioural activities.

## Results

**Genomic landscape of the G4 motifs in the genomes of the representative species at the evolutionary nodes.** The total number of the predicted G4 motifs ranged from 4705 in yeast to 1,323,932 in purple sea urchin (*Strongylocentrotus purpuratus*) and to 7,356,494 in humans (Supplementary Table 1), suggesting that during the large-scale evolution of organisms, the number of predicted G4s increased. Density of G4 motifs and the length ratio over the whole genome were further analysed. The analyses showed that as species evolved (Fig. 1a), the G4 motif density and length proportion to the whole genomes increased (Fig. 1b, c; Supplementary Table 1), suggesting that although the increase in the number of G4s in the genomes was partially associated to the increase in genome size, the increase in the number and density of G4s was due to the increase in species complexity. Genome simulation test further supported this suggestion (Supplementary Fig. 1). We generated a random sequence with the same genome size of each species and found that G4 number was generally increased along the set of these random sequences (Supplementary Fig. 1a) but the G4 density did not varied obviously, in despite of one exception (Supplementary Fig. 1b).

When focusing on the $(G/C)_3L_{1-7}$ motif, a representative construct of G4 motifs with well-demonstrated functional importance[6], a similar pattern of G4 motif evolution was also detected (Fig. 1e, f). Linear regressions for all of these trends consistently indicated a significant positive relationship between species evolution and G4 evolution. Notably, there were some exceptions in which the G4 density in individual invertebrate genomes, such as some species of Platyhelminthes and Coelenterata, was higher than that in mammalian genomes (Fig. 1b, c, e, f; Supplementary Table 1). One possibility is that the species in these metazoan progenitor groups might have complex and unique genomic features that are similar to those of vertebrates, including humans[34].

When the proportion of genes bearing G4 motifs in their upstream regulatory region was investigated throughout whole genomes, an obvious increasing pattern in the predicted G4 motifs was found (Fig. 1d, g), especially for the stable $(G/C)_3L_{1-7}$-type motif (Fig. 1g), with the higher fitness ($R^2$) in trend lines (Fig. 1). With the evolution of species from unicellular eukaryotic yeast to higher vertebrate humans, the distribution of G4 motifs in the genomes exhibited an increasing trend, and the number of G4 motifs intensively localized in the upstream regulatory regions of genes increased, reflecting that the role of G4, especially the most stable $(G/C)_3L_{1-7}$ type, in gene regulation may have become increasingly important during evolution.

To test the influence of genomic GC content on the emergence of G4 motifs, we plotted the GC content of each species. We found that the genomic GC percentage content basically remains stable during species evolution (Fig. 2a). However, when normalized with GC contents, the density of total G4 motifs (Fig. 2b) and $(G/C)_3L_{1-7}$ motifs (Fig. 2c) showed a general increase along the large-scale evolution. Thus, the effect of the genomic GC content seems not a major factor contributing to the observed increase in G4s in the genomes during species evolution.

We further demonstrated the distribution of the stable $(G/C)_3L_{1-7}$ motif in the chromosomes of nine representative species, yeast, dictyostelium, schistosomes, nematodes, fruit flies, zebrafish, lizards, chickens, and humans (Fig. 2d). The chromosome distribution of the $(G/C)_3L_{1-7}$ motif in the early organisms, such as *D. discoideum* and *S. mansoni*, was few, dispersed, and generally even. In the species that emerged later, such as nematodes, fruit flies, zebrafish, and lizards, the distribution of G4 motif in the chromosomes became more dense and intense. An increase in G4 motifs occurred in birds and mammals, such as red jungle fowl (*Gallus gallus*) and humans, and the distribution

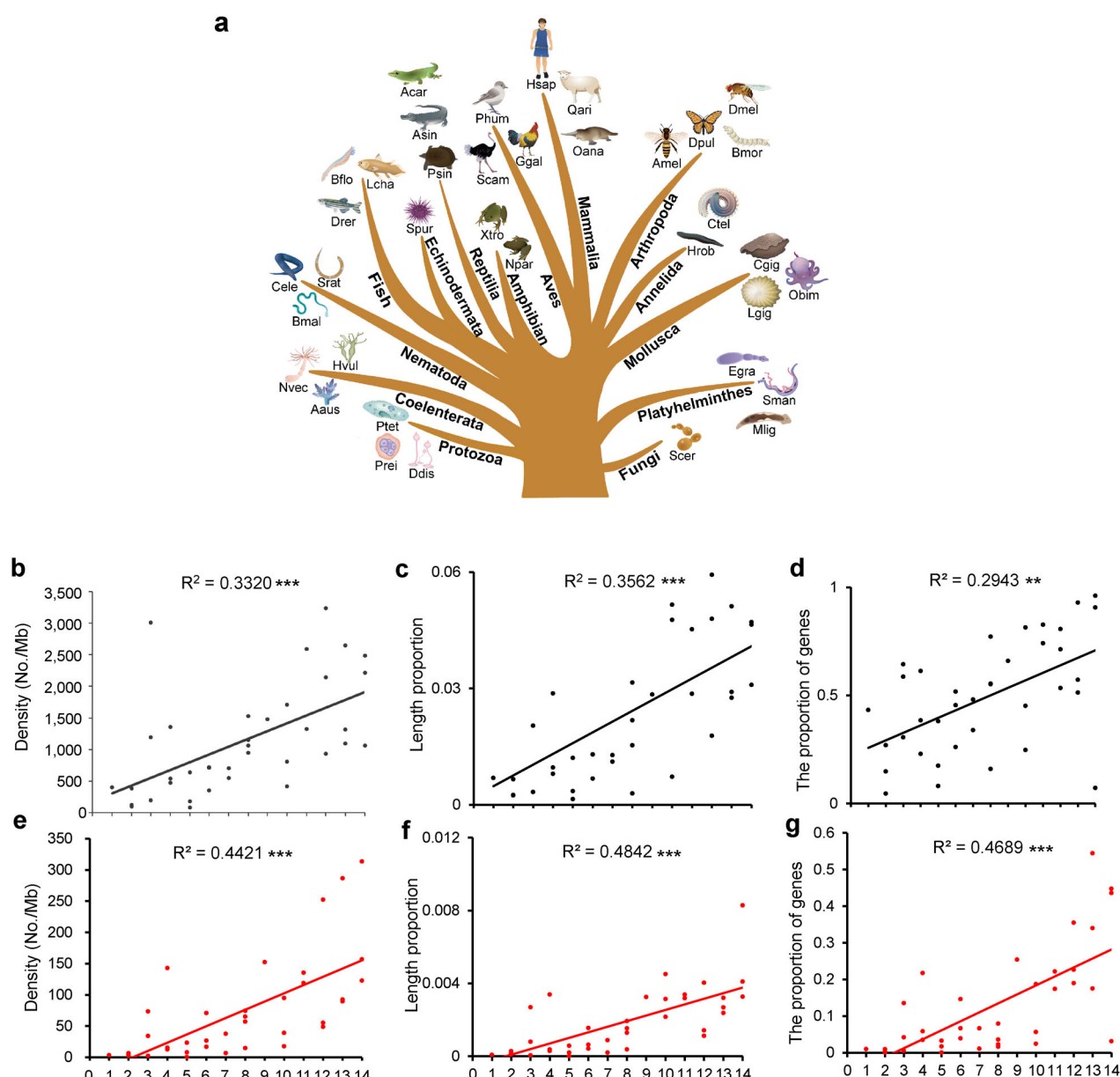

**Fig. 1 Genomic landscape of G4 motifs in the selected species of the representative phylogenetic nodes in the tree of life. a** Thirty-seven species in the phylogenetic tree, indicating the 14 representative evolutionary nodes from primitive eukaryotes to Animalia. **b**–**g** Genomic landscape of G4 motifs in the 37 species. The G4 density (**b**, **e**) and length ratio (**c**, **f**) in the genomes, and the ratios of the genes bearing G4 motifs in the upstream 2 kb region (**d**, **g**). **b**–**d** The data are presented based on all types of G4s [$(G/C)_2L_{1-4}$, $(G/C)_3L_{1-7}$, $(G/C)_3L_{1-12}$, and $(G/C)4L_{1-12}$, and $(G/C)_5L_{1-12}$]. **d**–**f** The data are presented based on the stable G4 structure $(G/C)_3L_{1-7}$. $R^2$: goodness of fit of the trend lines. $*p < 0.05$; $**p < 0.01$; and $***p < 0.001$ by F test. $n = 37$ biologically independent samples. Scer *Saccharomyces cerevisiae*, Prei *Plasmodium reichenowi*, Ptet *Paramecium tetraurelia*, Ddis *Dictyostelium discoideum*, Sman *Schistosoma mansoni*, Mlig *Macrostomum lignano*, Egra *Echinococcus granulosus*, Nvec *Nematostella vectensis*, Adig *Acropora digitifera*, Hvul *Hydra vulgaris*, Cele *Caenorhabditis elegans*, Srat *Strongyloides ratti*, Bmal *Brugia malayi*, Lgig *Lottia gigantea*, Obim *Octopus bimaculoides*, Cgig *Crassostrea gigas*, Ctel *Capitella teleta*, Hrob *Helobdella robusta*, Amel *Apis mellifera*, Bmor *Bombyx mori*, Dmel *Drosophila melanogaster*, Dpul *Danaus plex*, Spur *Strongylocentrotus purpuratus*, Lcha *Latimeria chalumnae*, Bflo *Branchiostoma floridae*, Drer *Danio rerio*, Xtro *Xenopus tropicalis*, Npar *Nanorana parkeri*, Acar *Anolis carolinensis*, Psin *Pelodiscus sinensis*, Asin *Alligator sinensis*, Ggal *Gallus gallus*, Phum *Pseudopodoces humilis*, Scam *Struthio camelus*, Oana *Octopus anatinus*, Oari *Ovis aries*, Hsap *Homo sapiens*. The numbers in **a** and the X-coordinate axes of **b**–**g** indicate the following phylogenetic categories: (1) fungus (Scer); (2) protozoa (Prei, Pter, Ddis); (3) Platyhelminthes (Sman, Mlig, Egra); (4) Coelenterata (Nvec, Aaus, Hvul); (5) Nematoda (Cele, Srat, Bmal); (6) Mollusca (Lgig, Obim, Cgig); (7) Annelida (Ctel, Hrob); (8) Arthropoda (Amel, Bmor, Dmel, Dpul); (9) Echinodermata (Spur); (10) Fish (Bflo, Drer, Lhal); (11) Amphibian (Xtro, Npar); (12) Reptilia (Acar, Psin, Asin); (13) Aves (Ggal, Phum, Scam); and (14) Mammalia (Hsap, Oana, Oari).

in birds and mammals was much denser and more obviously clustered than that in other species. In these two species with high G4 density, the G4s are not evenly distributed in different chromosomes and the high-density peaks are somewhat mosaic in a chromosome and different chromosomes. (Fig. 2d).

Furthermore, although G4 clusters in the red jungle fowl genome were quite dense, the density of G4s was higher in humans than in red jungle fowl (Fig. 2d). We further analysed the regions in the human genome with the highest (5%) density of the (G/C)₃L₁₋₇ motif and found that they were significantly enriched in

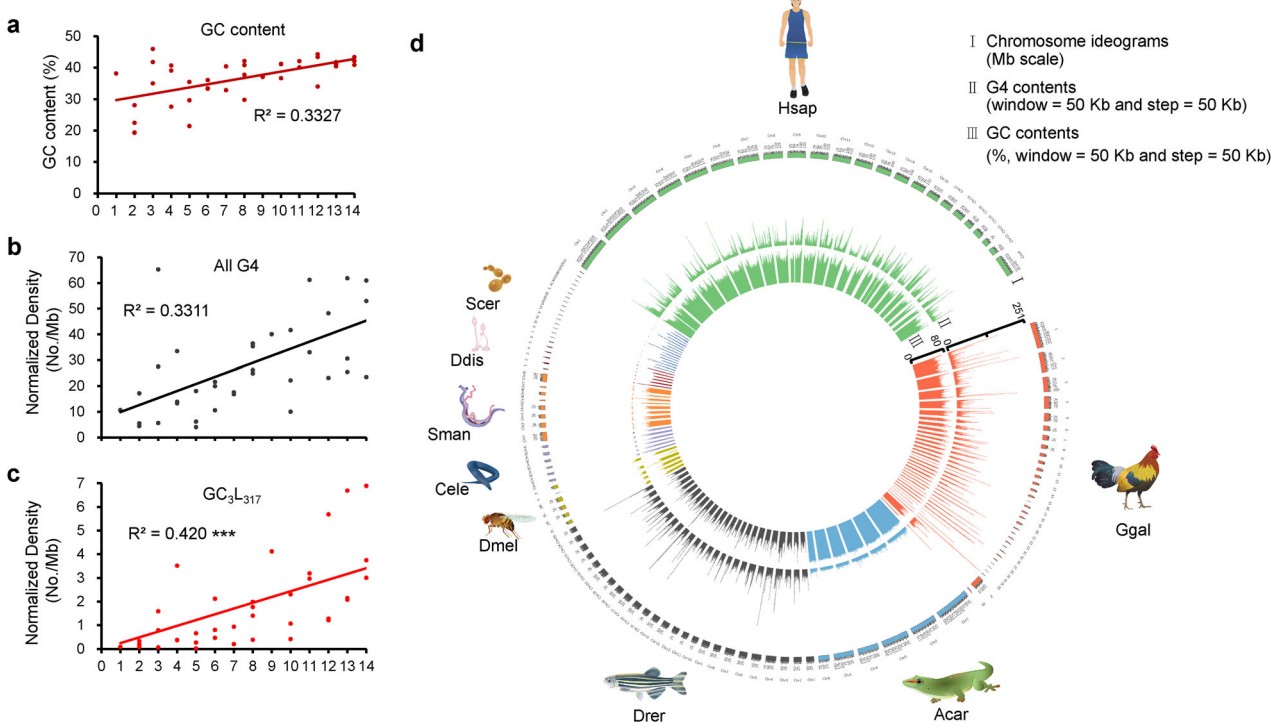

**Fig. 2 GC content and chromosomal distribution of the (G/C)$_3$L$_{1-7}$ G4 motifs in the selected species. a** The GC content of the genomes of the 37 species. **b, c** The density of total G4 motifs and (G/C)$_3$L$_{1-7}$ motifs, normalized over the GC contents. ***$p < 0.001$ by $F$ test. $n = 37$ biologically independent samples. **d** Chromosomic distribution of the (G/C)$_3$L$_{1-7}$ G4 motifs in nine selected species each from representing phylogenetic categories ranging from the fungus to mammal. The density of the (G/C)$_3$L$_{1-7}$ G4 motifs was calculated in a 50 kb sliding window with a 50 kb step and plotted in histograms along with each of the chromosomes of each species, indicating that G4 motifs are generally evenly distributed in the genomes of those species with low level of G4s, but clustered in those with high level of G4s (Ggal and Hsap), with high-density windows being separated by low-density windows. Histograms in different colours indicate the G4 motifs in different species. The abbreviation of each species is listed in the legend of Fig. 1.

**Table 1 Enrichment of the (G/C)$_3$L$_{1-7}$ G4 motif clusters in the genic regions of the human genome.**

| Functional region | Top 5% windows (3031) | Total windows (60,630) | $p$ value[a] |
|---|---|---|---|
| Upstream 2 kb | 2232 | 24,015 | <2.2e−16 |
| Gene body | 2763 | 39,184 | <2.2e−16 |
| Downstream 2 kb | 2284 | 24,082 | <2.2e−16 |

[a]$p$ values were calculated by chi-square test.

the gene body and the 2 kb upstream and downstream flaking regions of the genes (Table 1).

**Immunofluorescence detection of G4 structures in representative species.** Bioinformatics analyses revealed that the number and density of G4 motifs in genomes increased with the evolution of species. To further confirm this finding, immunofluorescence staining of G4 structures in cells of the seven selected representative species, *Saccharomyces cerevisiae*, *Drosophila melanogaster*, *Danio rerio*, *Pelodiscus sinensis*, *G. gallus*, *Ovis aries*, and *Homo sapiens*, was conducted using a well-known G4 antibody, BG4[35], which is fused with a His tag. Western blot confirmed that the anti-His antibody (Ab00174-1.6, absolute antibody) could specifically recognize BG4-His but could not recognize other proteins in the cells (Supplementary Fig. 2). Immunofluorescence staining showed that the fluorescence signals were very weak and barely detectable in most yeast

protoplasts (Fig. 3a). The fluorescence signals in *D. melanogaster*, *D. rerio*, *P. sinensis*, *G. gallus*, and *O. aries* increased gradually (Fig. 3b–f), and strong fluorescence signals were detected in human cells (Fig. 3g), consistently indicating an increasing trend during species evolution. The statistical data of fluorescence signals in these seven species verified that, from yeast to human, the number of G4 structures in the nuclei increased with the species evolution (Fig. 3h). As a control, there were no fluorescence signals in the nuclei of cells incubated without the BG4 antibody and the DNase I treatment significantly decreased the number of the foci in the nuclei (Fig. 3a–h).

**Functional classification of the genes bearing the G4 motif in upstream regulatory regions.** Considering the marked increase in G4 motifs in genes, particularly in regulatory regions with organism evolution, we further analysed the functional enrichment of genes bearing G4 motifs in the upstream 2 kb regulatory regions in 19 species that have Gene Ontology (GO) annotation of the gene set (Supplementary Data 2). In the majority of these species, the proportions of transcription factor genes in (G/C)$_3$L$_{1-7}$ motif-bearing genes were obviously higher than those in the background (the proportions of transcription factor genes in all the genes) (Fig. 4a), implying an aggregation effect of G4 motifs in transcription factor genes. Notably, in the phylogenetic groups that evolved later, starting from insects, transcription factor genes were significantly enriched in the G4 motif-bearing genes (Fig. 4a, Supplementary Table 2). These results suggest that with the species evolution, G4 motifs tended to be enriched in the promoter regions of genes, particularly those genes with transcription factor activity.

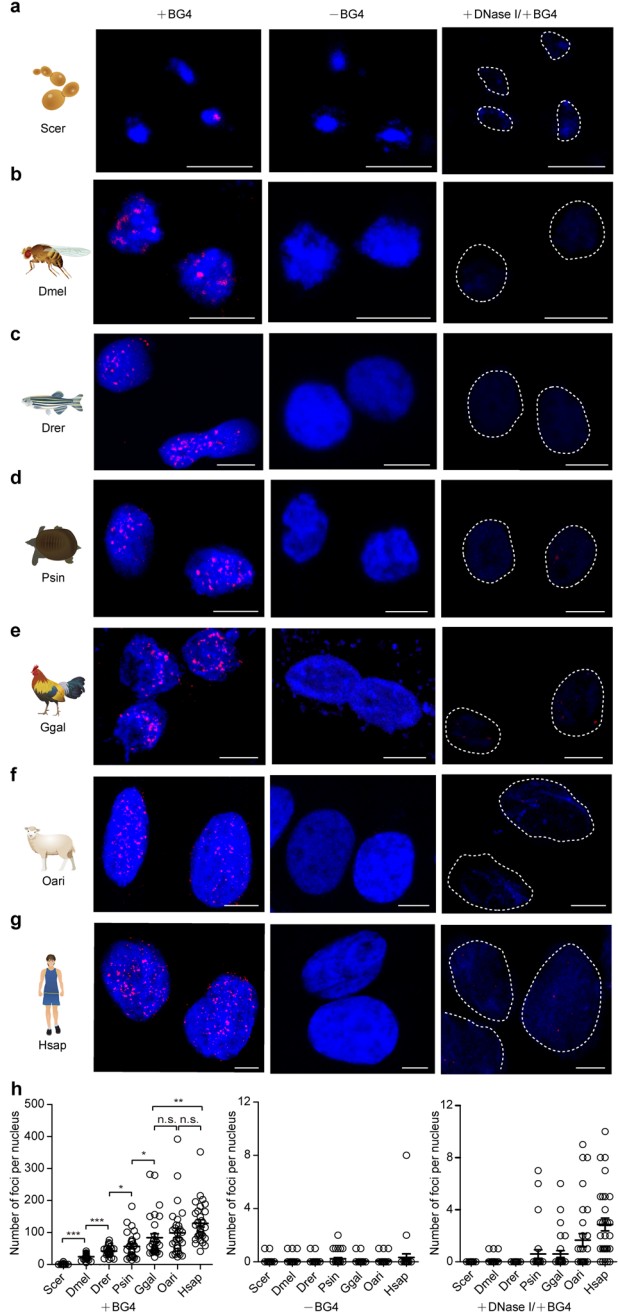

**Fig. 3 Detection of the G4 structures in the representative species by immunofluorescence staining. a–g** Immunofluorescence staining signals showing the G4 structures in *Saccharomyces cerevisiae* (Scer), *Drosophila melanogaster* (Dmel), *Danio rerio* (Drer), *Pelodiscus sinensis* (Psin), *Gallus gallus* (Ggal), *Ovis aries* (Oari), and *Homo sapiens* (Hsap). **h** Quantification statistics of the number of G4 structures per nucleus in Scer, Dmel, Drer, Psin, Ggal, Oari, and Hsap cells. Signal spots in 30 nuclei from three replicates (ten nuclei for each replicate) were counted for each species. Data are the mean ± SEM ($n = 30$), statistical significances were determined by Student's *t* test, *$p < 0.05$, **$p < 0.01$, and ***$p < 0.001$. The scale bars equal 5 μm.

### Analysis of the G4 motif in orthologous genes.

The aggregation of the $(G/C)_3L_{1-7}$ G4 motif in the upstream region of the genes implies the increasing importance of the G4 structure in gene regulation during the large-scale species evolution. Whether this aggregation occurs parallelly in orthologous genes among different species is an interesting concept. We tested this hypothesis

in one representative species in each of the 14 phylogenetic nodes (Fig. 4b). With the reciprocal BLAST method with *Drosophila* as the core species, 1467 (Dmel–Prei)–5871 (Dmel–Ctel) pairwise orthologous genes between *D. melanogaster* and the corresponding species were identified (Supplementary Table 3). In total, 720 common orthologues were identified in each of the 14 species. For both the pairwise and common orthologous genes, the proportion of the genes containing G4 motifs in the upstream 2 kb region generally increased with evolution in this large-scale analysis (Fig. 4b). Birds and mammals showed the highest proportion of genes bearing G4 (Fig. 4b).

**G4 loop-length distribution**. The lengths of the three loops in a G4 structure have an important influence on the stability of the structure[36]. The genomic landscape of the loop-length distribution of the stable $(G/C)_3L_{1-7}$ motif in species evolution was analysed with a focus on the G4 motifs in the upstream 2 kb regions of the genes in all representative species. As shown in Fig. 5a, during evolution, the diversity of the loop-length types generally increased. In those primary groups, such as fungus and protozoa, $(G/C)_3L_{1-7}$ motifs were limited to a few loop-length types. New types of motifs emerged in the genomes of invertebrates that evolved later, but the frequencies of diverse loop-length motifs were still low (Fig. 5a). However, in vertebrates, the frequencies of the diverse loop-length G4 motifs significantly increased. These results suggest that the loop-length patterns became diverse with evolution, resulting in an overall increase in the number, type, and complexity of G4 motifs, which may benefit the fine regulation of gene transcription in higher organisms.

Hierarchical clustering of the frequencies of the different loop lengths of the $(G/C)_3L_{1-7}$ motifs revealed a clade of G4 motifs with a high frequency (Fig. 5a, b). These motifs had relatively shorter loop lengths, especially in the second loop (Fig. 5b). Among them, the 1:1:1 type (length of the first, second, and third loop is all one base) showed a markedly high frequency in the majority of species (Fig. 5b). This type of motif was within the top ten most frequently occurring motifs in the majority of the species (31 out of all 37 species) (Fig. 5c).

**Relationship between G4 motifs and methylation**. DNA methylation and G4 structures both occur in GC-rich regions. To investigate the relationship between these two structural characteristics, a mammalian species with a highly methylated genome, the pig, and a species with relatively low methylation, the silkworm, were used to examine the possible evolutionary relationship between G4 motifs and methylation in the upstream 2 kb regions of genes. Generally, in the highly methylated pig genome, cytosines in the upstream 2 kb regions of the genes bearing $(G/C)_3L_{1-7}$ showed significantly lower methylation levels than those in the whole gene set (Fig. 6a). We further plotted the methylation levels in 200 bp sliding windows in the upstream 2 kb regions of genes and found that in each 200 bp window, the methylation level of the $(G/C)_3L_{1-7}$-bearing genes was obviously lower than that in the whole gene set background (Fig. 6b). In the sparsely methylated silkworm genome, the frequencies of methylated cytosines (mCs) were significantly lower in the upstream region of $(G/C)_3L_{1-7}$-bearing genes than in all genes (Fig. 6c). When the methylation level of these mCs was examined, the $(G/C)_3L_{1-7}$-bearing genes had significantly lower frequencies of hypermethylated cytosines than all genes (Fig. 6d). These results suggest that an antagonistic relationship between DNA G4s and methylation may exist and may be conserved, at least in mammals and insects.

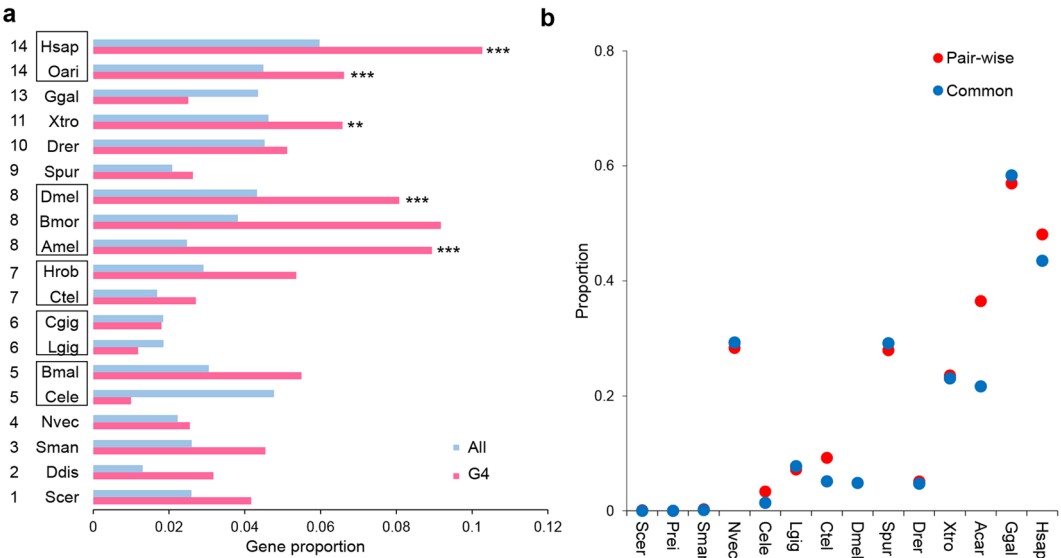

**Fig. 4 Functional landscape of the (G/C)₃L₁₋₇ G4 motifs during evolution. a** The proportion and enrichment of the genes bearing the $(G/C)_3L_{1-7}$ G4 motif in the upstream 2 kb regions with transcription factor activity. The red histogram indicates the proportion of transcription factors in the genes bearing the $(G/C)_3L_{1-7}$ G4 motifs in their upstream 2 kb region. The blue histogram indicates the proportion of transcription factors in all gene sets. The abbreviation of each species name and the numbers representing phylogenetic nodes are listed in the legend of Fig. 1 Histograms marked with stars indicate the significance of the enriched molecular function of the transcription factor activity by GO enrichment analysis. $**p < 0.01$ and $***p < 0.001$ by hypergeometric test with false discovery rate correction. Detail information of GO enrichment results were shown in Supplementary Data 2 and Supplementary Table 2. **b** Proportion of genes bearing the $(G/C)_3L_{1-7}$ G4 in the upstream 2 kb regions in the orthologues among different species. Pairwise: pairwise orthologues between *Drosophila melanogaster* and each of the 12 species listed in the X-coordinate. Common: orthologues common in all 13 species.

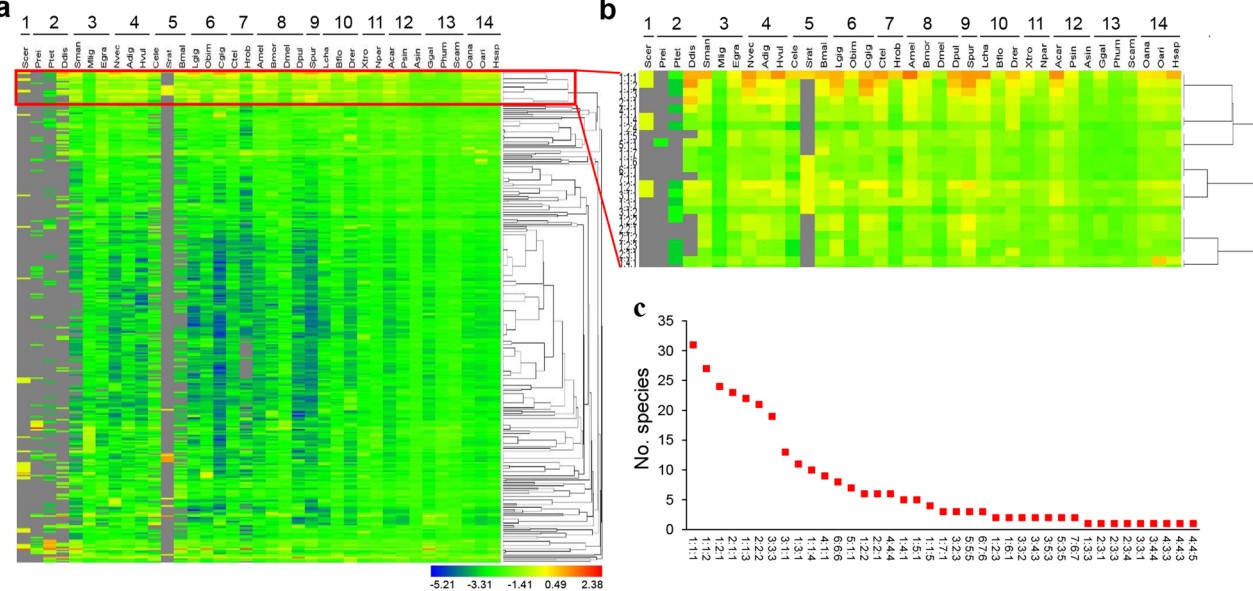

**Fig. 5 Loop lengths of the (G/C)₃L₁₋₇ G4 motifs in the genomes of the 37 representative species. a** Heat map presentation and hierarchical clustering of the proportion of the $(G/C)_3L_{1-7}$ motifs with each of the loop-length types. **b** Zoomed out view of one cluster in **a**, showing the G4 motifs with a relatively higher proportion in the majority of the species. **c** Species communality of the top ten dominant loop-length types of the $(G/C)_3L_{1-7}$ motifs in each species. The Y-axis shows the number of species that have the corresponding loop-length type of the $(G/C)_3L_{1-7}$ motifs in their top ten motifs. Numbers spaced by colons indicate the base length of the first, second, and third loops, respectively. The abbreviation of species is listed in the legend of Fig. 1.

## Discussion

In this study, large-scale evolutionary patterns of the genomic G4 motifs were analysed for selected representative species of organisms from single-celled eukaryotic fungi to humans by using the classical algorithm quadruplexes[37]. Recently, a lot of newly developed tools based on different algorithms have been used to predict new types of quadruplexes, which are beyond the prediction by quadparser[38]. The limitations of quadparser probably is its highly stringency in prediction of G4, which may cause false negative. It may not efficiently detect some imperfect or noncanonical quadruplexes. Thus, it may not be sufficient for the preliminary and reliable screening of low-abundant potential

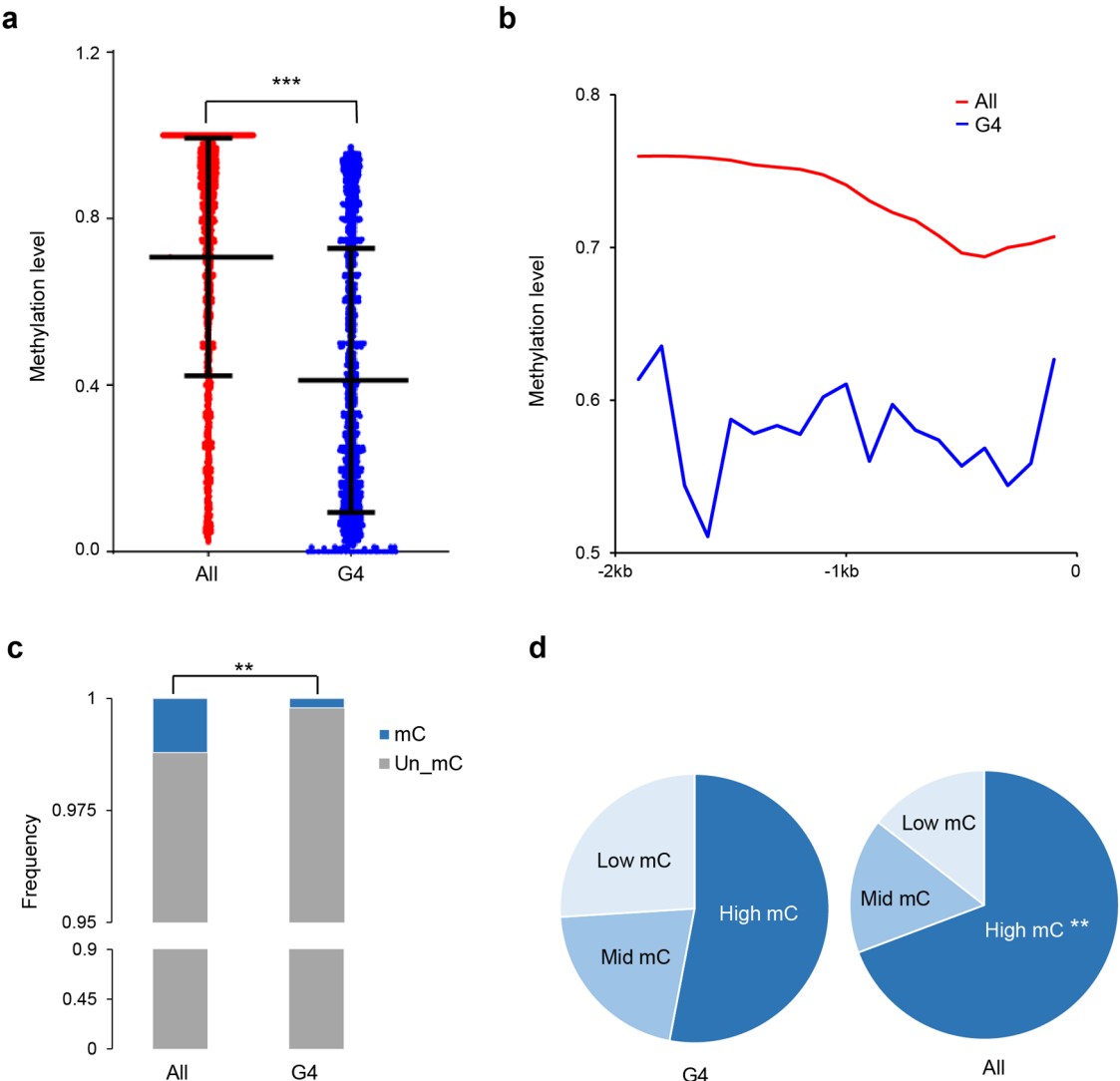

**Fig. 6 Relationship between the DNA G4 motifs and CpG methylation level in the upstream 2 kb regulatory regions of genes in pig and silkworm. a** The methylation levels of the cytosines at upstream 2 kb region of all genes (All) and those genes bearing the $(G/C)_3L_{1-7}$ motifs in their upstream 2 kb region (G4) in the pig. Bars from the top to the bottom indicate the value of the three quarters, the median and a quarter respectively. ***$p < 0.001$ by Wilcoxon test. $n = 34,333$ cytosine sites for the former and 148,769 cytosine sites for the latter. **b** Plot of the methylation level distribution in the upstream 2 kb regulatory region of genes in the pig. Methylation levels were calculated in 200 bp sliding windows with a 100 bp step. Proportion of unmethylated and methylated cytosines (**c**) and the methylated cytosines (mCs) with different methylation levels (**d**) in the upstream 2 kb region of all genes (All) and those genes bearing the $(G/C)_3L_{1-7}$ motifs in their upstream 2 kb region (G4) in the silkworm. High mC hypermethylated cytosines (methylation level > 0.6), mid mC moderate methylated cytosines (0.3 < methylation level < 0.6), low mC hypomethylated cytosines (0 < methylation level < 0.3). **$p < 0.01$ by chi-square's test (data subjected to this test were shown in Supplementary Data 1). All: the results of all genes; and G4: the results of the genes bearing the $(G/C)_3L_{1-7}$ motifs in their upstream 2 kb region.

conformations or sequences by computational analysis. In this study, because our core interest is the evolutionary pattern of G4s, the stable and credible type of G4 motifs, i.e., $(G/C)_3L_{1-7}$, was used to predict. The types of G4 motifs with longer loop length, such as loop1–12, were also analysed. The results indicated that G4 motifs with longer loop length showed similar genomic distribution pattern with $(G/C)_3L_{1-7}$ type, (Supplementary Fig. 3). A series of genomic characteristics, including the absolute number and density of G4s (Fig. 1b, e), the proportion of genes that contain G4s to all genes (Fig. 1d, g), functional enrichments of G4-bearing genes (Fig. 4a), and immunofluorescence detection of G4s (Fig. 3), demonstrated that during the large-scale species evolution, the G4 structures, especially the most stable $(G/C)_3L_{1-7}$ type, gradually increased in genomes, not only in number but also in density and diversity. The genomes of lower organisms are

simpler than those of higher organisms. With the increase in the complexity of animal physiology and behaviours, the genome sizes of organisms have enlarged. However, the GC content is generally maintained at a relatively conserved level (Fig. 2a). Under this circumstance, an increase in the GC-rich G4 motifs appears not simply because of the GC content. More importantly, the increase in the number and density of the G4 motifs in the genomes is not simply due to the increase in genome size. At some unique stage in the evolution process, for example, the metazoan progenitor stages of Platyhelminthes and Coelenterata, the number and density of the G4 structures were not proportionally associated with the increase in genome size (Fig. 1, Supplementary Table 1). Further, we generated a random sequence with the same genome size of each species and found that G4 number was generally increased along the set of these

random sequences (Supplementary Fig. 1a) but the G4 density did not varied obviously, in despite of one exception (Supplementary Fig. 1b).

Organisms appear to develop more G4 structures in the genic regulatory regions of genes than in other regions[6], and the distribution of the G4 motifs in chromosomes and genomes appears not to be random, with more G4 motifs in the regulatory regions of genes than in other regions and sometimes with G4 clusters in specific chromosomes and regions. During evolution, the G4 structures appear to have become more enriched in transcription factor genes than in other genes (Fig. 4a), which would facilitate the elaborate regulation of gene expression in cascade signal transduction. These characteristics suggest that the DNA secondary structures have probably evolved into a new and important epigenetic mechanism for gene regulation to meet the multiple requirements for physiological and behavioural demands in higher organisms. Similar results have also been found in other small-scale studies. For example, the density of G4 in the transcription start site ±5k was found to be much higher in mammals such as humans, chimpanzees, and mice than in fruit flies and nematodes[29,39]. During species evolution, the G4 motifs probably evolved from a sparse distribution in the primary phylogenetic groups to a densely clustered distribution in mammals. G4 clusters are evident in humans but not in the Schistosoma and nematode genomes and are sparse in the chromosome of single-celled yeast[28]. In addition, the highly dense G4 clusters in humans are enriched in the regulatory regions of genes, which was also observed in a previous study[6]. These results consistently imply the increased functional impact of G4 structures on gene transcription regulation. Thus, a regulatory function of G4 structures might not have been established in lower organisms until their development in higher animals, where they developed into a new epigenetic mechanism by forming a large amount of G4 structures.

Given the increasing pattern of G4s in the genome during species evolution, the stability of the G4 structures in the genomes of different species is an interesting issue because it is associated with the persistence and function of these structures in genomes. The loop length, especially the central loop length in G4 motifs, has a critical influence on the stability of the resultant quadruplexes. A short loop, particularly a short central loop length, facilitates a stable G4 conformation[36,40]. In humans, the G4 motifs with the shortest loop length (for example, 1:1:1) construct the most stable structures in the genome[24]. In this study, except for the two most basal phylogenetic groups, fungus and protozoa, this type of G4 motif with the shortest loop length was the most conserved during species evolution. This structural conservation pattern also implies the functional conservation of the G4 motifs in diverse species. On the other hand, the diversity of the loop-length types was also found to increase, especially in the groups that emerged later. The significance of the diverse types of G4 motifs is still pending further exploration. One possibility is that the increasing dynamics of the DNA secondary structures may provide additional opportunities for the fine regulation of gene transcription.

One of the interesting findings in this study was the increased enrichment of G4-bearing genes in transcription factors, which was first detected in the silkworm[41]. Previous studies attempting to explore the functional implication of G4-bearing genes suggested that these genes may be involved in many processes, such as carcinogenesis in humans[27], drought stress-related metabolic processes in *Arabidopsis*[42], processes related to energy status, hypoxia, low sugar, and nutrient deprivation in maize[19] and development, cell growth, and transmembrane transport processes in monocotyledonous and dicotyledonous plants[20]. Establishing a functional relationship between G4 structures and transcription factors is critically necessary because the regulation of transcription factors is more or less directly and extensively involved in multiple biological activities, considering that G4 structures have been evolved into a novel regulatory mechanism of gene transcription. It should be noted that genome annotation reliability are largely affected by the quality of genome assembly. In our study, we tried to select those species with high-quality sequenced genomes. We cannot guarantee all the genome sequences reported by other laboratories are perfectly or totally accurate, but we believe that the overall trend of the analysis results is reliable. Actually, we could not exclude the any possible mistakes in genome assembly and gene annotation by other groups.

Notably, the above findings were based on sequence prediction by using the sequence model of G4 motifs. The predicted numbers and density do not necessarily guarantee the actual formation and persistence of these structures inside cells. Whether and when these structures form in the cell nuclei of organisms needs to be experimentally demonstrated and confirmed. In this study, cells of seven representative species from lower to higher organisms were examined for the in vivo existence of G4 structures by immunofluorescence staining using a specific anti-G4 antibody, which recognizes the structure itself, not the sequence[35]. The results clearly confirmed the findings from the sequence prediction. During the large-scale species evolution from yeast to human, the signals for G4 structures increased significantly (Fig. 3). Thus, the evolutionary tendency of G4 structures is confirmed not only by sequence prediction but also by direct structure detection. It is believed that the formation or destruction of G4 secondary structures in cells depends on cell physiological status, such as iron concentration, pH and other factors in cell nuclei[43]. However, this reversible characteristic of the G4 structure allows it to become an elaborate regulatory mechanism of gene transcription.

Gene transcriptional regulation could be achieved through genetic and epigenetic systems. DNA methylation is an epigenetic mechanism and usually occurs in the GC-rich regions of the genomes, where DNA secondary structures also are formed. The relationship between these two epigenetic mechanisms in gene transcription regulation remains unclear. In humans, G4 has been reported to antagonize DNA methylation[31,33,44]. In this study, the evolutionary relationship between G4 motifs and DNA methylation was examined using two phylogenetic animal models. The results indicated that in regions with a high density of G4 clusters, the DNA methylation rate was low and vice versa. Furthermore, this antagonistic occurrence between G4 structures and DNA methylation may have occurred as early as the emergence of insects and may have been maintained for the subsequent evolution post-insect emergence.

In summary, during the long-term evolution of organisms, DNA secondary structures appeared and gradually evolved in genomes. The number and distribution density of the structures from lower to higher organisms exclusively increased, probably facilitating the development of a new method of gene regulation, which does not depend on the primary sequence of the genomes. This epigenetic regulatory mechanism would aid higher organisms in reversibly and elaborately regulating gene transcription to achieve increasingly complex cellular, physiological, and behavioural activities.

## Methods

**Selection of species and genome data**. A total of 37 species from fungi to mammals in the 14 representative phylogenetic nodes from the primitive eukaryotes to animalia were selected to perform a large evolutionary scale analysis. The phylogenetic tree was generated by referring the well-recognized evolutionary pattern[37]. At each of the phylogenetic nodes, two or three species with different evolutionary statuses and available high-quality genome information were selected from 14 categories: fungi, protozoans, Platyhelminthes, coelenterates, nematodes,

molluscs, annelids, arthropods, echinoderms, fish, amphibians, reptiles, birds, and mammals (Fig. 1a, Supplementary Table 4). Genome sequences, protein sequences, and annotation files of all of these selected species were obtained from Ensembl, NCBI, and SilkDB. The links to download the detailed genome and annotation information are shown in Supplementary Table 4. The relevant GO annotation data were obtained from the Ensembl database (http://asia.ensembl.org/biomart/martview/a711e156a54e647e61290eadf58122a5).

Random stimulated sequence with the same genome size of each species was generated by an in-house Perl script.

**Genomic G4 motif analyses**. Genomic screening for G4 motifs was conducted using the quadparser algorithm[24]. There are many types of G4 structures. The number of Gs in G-tract can be 2, 3, 4, or 5[38,45] and the loop length usually ranges from 1 to 7 or even longer[38]. The stability of G4s usually decreases with longer loop length[27]. When two Gs in a G-tract exist, the more stable G4 structure tends to have the loop with length of 1–4 bp[45]. Therefore, the types of potential G4 motifs were included: $(G/C)_2L_{1-4}$, $(G/C)_3L_{1-7}$ and $(G/C)_3L_{1-12}$, $(G/C)_4L_{1-12}$, and $(G/C)_5L_{1-12}$ (here, L represents the loop length of the G4 structures; the numbers represent the number of bases)[46]. The density and length ratio of the G4 motifs were determined by the number of G4s and the total length of G4s, respectively, divided by the total length of the genome. For each plot, a trend line was fitted with linear regression. In this study, we considered the complementary sequence of G4, i.e., the C-rich region in the genome sequence (the plus stand), as the G4 motif in the minus strand. To normalizing the number and density of G4s in the genomes with different GC contents, the number and density of G4 were divided by the GC content for each species.

**Analyses of the $(G/C)_3L_{1-7}$-type G4 motifs**. The G4 distribution patterns of all types of G4 motifs among the species were consistent with those of the $(G/C)_3L_{1-7}$ type, which is the most stable structure with well-supported functionality by experimental evidence[6]. Therefore, this type of G4 motif was the focus of further analyses. The chromosomal density distribution of this type of G4 motif was calculated as the total number of G4s per 50 kb genomic window with a 25 kb step. CG content in each corresponding window was also calculated. Genomic plotting of the G4 density distribution and CG content were generated using Circos[47]. The functional distribution of G4 motifs in the genomes was determined by overlapping G4s with the upstream 2 kb regions and gene bodies.

**Identification of orthologous gene sets**. One species in each phylogenetic category was selected for identification of orthologous gene. Protein sequences of *D. melanogaster* were used as a core gene set for searching pairwise orthologous genes with the other 13 species by using reciprocal BLAST (E-07)[48]. When core genes had pairwise orthologues in each of the 13 species, this set of genes was regarded as common orthologues.

**GO enrichment analysis**. The relevant GO annotation data were obtained from BioMart (http://asia.ensembl.org/biomart/martview/a711e156a54e647e61290eadf58122a5) in the Ensembl database. The functional enrichment analysis was conducted using an online platform OmicShare (http://www.omicshare.com/tools/Home/Soft/gogsea) with hypergeometric test, followed by false discovery rate correction.

**$(G/C)_3L_{1-7}$ loop-length distribution**. There are three loops in each of the G4 motifs. The length of each loop in each motif was calculated by quadparser[24]. In total, the G4 motifs were divided into 342 loop types (from 1:1:1 to 7:7:6)[20].

**Relationship between $(G/C)_3L_{1-7}$ and DNA methylation**. A mammal, the domestic pig *Sus scrofa domesticus*, and an insect, the domesticated silkworm *Bombyx mori*, were used as subjects to initially explore the relationship between the $(G/C)_3L_{1-7}$ motif and DNA methylation. Data on DNA methylation profiling of the pig were provided by Li et al.[49] (raw data were downloaded from NCBI with the accession no. GSM2429533) and those of the silkworm were downloaded from ftp://ftp.genomics.org.cn/pub/silkworm_methylation. Methylation level of each cytosine site at upstream 2 kb region of each gene was calculated by the number of methylated reads divided by all the reads covering the site. In the pig, DNA methylation in the upstream 2 kb region of genes was calculated in 200 bp sliding windows with a 100 bp step.

**Immunostaining G4 structures in cell nuclei**. Seven representative species were used to examine the G4 structures in the genomes by using immunofluorescence staining with anti-G4 antibody (BG4-His, Absolute antibody, Oxford, UK). *D. melanogaster* KC cells, *G. gallus* DF-1 cells, *O. aries* OAR-L1 cells, and *Homo sapiens* LO2 cells were loaded onto glass coverslips and fixed in 2% paraformaldehyde for 30 min at 4 °C, permeabilized with 0.1% Triton-X100 at 4 °C for 30 min, and then blocked in blocking solution (5% bovine serum and 5% goat serum in PBS) overnight at 4 °C. After blocking, the cells were incubated with BG4-His antibody (1 μg/ml) in a 1:10 diluted blocking solution overnight at 4 °C. For the control, the anti-His antibody (B1023, Beijing Biodragon ImmunoTechnologies,

Beijing, China) was used at a 1:1000 dilution. On the following day, the cells were incubated with anti-rabbit Alexa 594-conjugated (A11037, Invitrogen, CA, USA) secondary antibodies at a 1:2000 dilution in a 1:10 diluted blocking solution for 1 h. The cells were washed with ice-cold PBS supplemented with 0.1% Tween 20 four times. Then, the coverslips were mounted with Prolong Gold/DAPI (P36941, Invitrogen, CA, USA), and confocal images were obtained under an Olympus Fluoview FV1000 confocal microscope. Protoplasts from *S. cerevisiae* were produced according to kit instructions for protoplast preparation (BB36301, BestBio, Shanghai, China). *D. rerio* cells and *P. sinensis* cells were produced from adult fish and *Pelodiscus* by using Trypsin-EDTA (25200072, Thermo Scientific, MA, USA) digestion. The *S. cerevisiae* protoplasts, *D. rerio* cells, and *P. sinensis* cells were fixed on poly-L-lysine (P4707, Sigma-Aldrich)-coated glass coverslips in a 12-well plate. The subsequent steps were identical to those described above. In the control group, the steps were the same as above except for the incubation of the BG4 antibody. *D. melanogaster* KC cells, *G. gallus* DF-1 cells, and *Homo sapiens* LO2 cells were obtained from the American Type Culture Collection. *O. aries* OAR-L1 cells were provided by Kunming Instituted of Zoology. The adult fish of *D. rerio* (male) were purchased from China Zebrafish Resource Center (CZRC Catalog ID: CF1). The *P. sinensis* were purchased from *P. sinensis* farmers.

**Western blot analysis**. For western blot analysis, total protein extracts from protoplasts of *S. cerevisiae*, *P. sinensis*, KC, LO2, DF-1, and OAR-L1 and *D. rerio* cells (30 μg) and BG4 (2 μg) were separated in 12% SDS-PAGE and transferred to a nitrocellulose membrane. The membrane was blocked with 3% bovine serum albumin in Tris-buffered saline with Tween (TBST) (10 mM Tris-HCl, pH 7.5, 150 mM NaCl, 0.05% Tween 20) for 2 h at room temperature and then incubated with the anti-His antibody in a 1:5000 dilution at 4 °C overnight. The samples were then incubated for 1 h at room temperature with the secondary antibody horseradish peroxidase-conjugated goat anti-rabbit IgG at a 1:7500 dilution (Dingguo Biotechnology, Guangzhou, China). Immunoreactivity was detected by enhanced chemiluminescence (Invitrogen, CA, USA). The membranes were washed with TBST three times, each for 5 min, before colour development.

**Statistics and reproducibility**. All the genomic data of the species are available with link formation listed in the "Methods" section or Supplementary Information files. Random simulated genome sequences could be repeatedly generated by the Perl script which will be provided upon request. All statistical methods were listed in the main text or in the figure legends. *F* test was conducted by the SPSS Statistics (Version 2.2). The samples are the 37 species. Wilcoxon test and chi-square test were conducted by RStudio (Version 3.6.1). The samples are the number of corresponding cytosine sties or G4 motif windows. Functional enrichment analyses were conducted using an online platform OmicShare (http://www.omicshare.com/tools/Home/Soft/gogsea) with hypergeometric test, followed by false discovery rate correction. The samples are corresponding genes. In immunofluorescence, signal spots in 30 nuclei from three replicates (ten nuclei for each replicate) were counted for each species. Data are the mean ± SEM ($n = 30$ biologically independent samples), statistical significances were determined by Student's *t* test.

**Reporting summary**. Further information on research design is available in the Nature Research Reporting Summary linked to this article.

## Data availability

The data that support the findings of this study are available from the authors on reasonable request. The source data underlying figures are presented in Supplementary Data 1.

## Code availability

Codes for predicting G4 motif sequences and loop-length pattern were according to the instruction of the C code quadparser (version 2.0, http://www-shankar.ch.cam.ac.uk/quadparser.html). Other codes used to analyse data are available at https://github.com/CuiYong01/G4-evolution.

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

## Acknowledgements

We thank Prof. Zhang Wang for his constructive suggestions in generating random simulated genome and Prof. Qing Wei for his help in material collection. This work was supported by grants from the National Natural Science Foundation of China (nos 31672494, 31720103916, 31930102) and China National Postdoctoral Program for Innovative Talents (BX20190123).

## Author contributions

Q.F. conceived, designed, and supervised the project. H.X. supervised the bioinformatic analyses. F.W., Y.C., and C.L. collected genomic and methylomic data and generated bioinformatic analyses. K.K.N. performed experiments and related analyses. M.L. generated phylogenetic tree of the thirty-seven species. Y.R. provided help in data collection and Perl-script writing. Y.F.C., H.D., L.H., S.Z., and L.L. helped with experimental material and methods. F.W., K.N., and H.X. wrote the manuscript with input from all authors. Q.F., Q.S., J.W., H.X., and K.N. revised the manuscript. All the authors reviewed and approved the paper.

## Competing interests

The authors declare no competing interests.
