## [Peer Review File · Communications Biology]

Reviewers' comments:

Reviewer #1 (Remarks to the Author):

In this manuscript, Wu et al perform a bioinformatic search for G4 motifs in 37 different species. This study suffers from the following shortcomings:

These 37 species are claimed to be "critical evolutionary nodes in the tree of life"... but there is not a single prokaryote or plant in this list! Why choose *Plasmodium reichenowi* and not *Plasmodium falciparum*? In addition, what is the evidence that these species are actually "representative"? Later in the ms, the authors select a subset of these 37 species (for example 9 in figure 2a) again claiming these are "representative species".

Table S1 - minus the last column - could be moved to the main MS. In this Table, the link to *Octopus anatinus* is wrong (the one listed corresponds to *Octopus bimaculoides*). The low number of genes in this species is surprising. How reliable are the numbers of genes provided for each species? 42,454 in humans vs only 2,717 in *Octopus anatinus*?

Also annoying are statements implying that the proportion of G4 bearing genes increased "with species evolution". A very simple organism can result from a highly sophisticated evolution process and should not be considered primitive.

Fig 1a: where does this tree of life come from? Provide a reference for the "general order of divergence during evolution". Is there a consensus on the "general order of divergence" proposed here?

The limitations of quadparser are not discussed – more accurate prediction tools are available. I strongly suggest the authors to read the guide to computational methods for G4 predictions recently published in NAR (doi: 10.1093/nar/gkz1097)

"All four types of potential G4 motifs were included" but i) one could define other types (longer loop lengths have been tested; see reference cited in the previous §) and ii) the interest of each of these 4 types is not explained. When counting the number of G4 motifs, how do the authors treat overlapping motifs (for example does (GGGTTA)₅ count for one or two quadruplexes?).

Genome annotation reliability (for gene definition, transcription factors) and even assembly (because of repetitive motifs) may differ between species – the authors never discuss this point.

Fig 2a: It would be more informative to plot the density of G4 motifs vs GC content for all species.

Fig 2b is useless at this magnification. What is the central circling arrow supposed to mean? Evolution time??

Fig 3: was a trivial control (adding DNase) performed to make sure these foci correspond to G4 DNA? Fig 3a is a rather trivial control that only needs to be showed in sup info. The problem to count BG4 foci number is that the fluorescent spots have different intensities. Was a threshold level defined?

Fig 6: Are pig and silkworm methylations pooled together? What is the fraction of G4 genes in both cases?

Fig 6c is misleading (and histogram starts at 0.95!). It would be more informative to plot the % of mC in both cases.

Table 2 is of little interest in the main MS – should be move to S.I.

Reviewer #2 (Remarks to the Author):

In this manuscript Wu et al study the presence of G-quadruplex structures (G4s) in 37 different species representative of the different evolutionary stages of the eukaryotic genome. By a bioinformatics approach they show that the abundance and complexity of G4s augmented with the evolution and with the increasing complexity of the considered eukaryotic organisms. They show the abundance of transcription factors among genes bearing G4s and that methylation sites are antagonized by the presence of G4-forming sequences. By immunofluorescence they visualized the abundance of G4 in 4 representative species.

Overall the findings are of general interest and further support the G4-mediated regulation of cellular genes in organisms. The manuscript is rationally organised and results clearly presented.

I would suggest addressing the following major and minor comments to improve the manuscript.

Major comments

1) The authors state: "All four types of potential G4 motifs were included: (G/C)₂L1-2, (G/C)₂L1-4, (G/C)₃L1-3 and (G/C)₃L1-7". However, there are many more potential types, with longer loops (as the authors also show) and with a higher number of G/C layers (e.g. 4). Please revise this throughout the text.

2) At page 7 the authors state: "...suggesting that the increase in the number of G4s in the genomes was not only because of the increase in genome size but also...". This is not correct because the increase in the genome size cannot be considered since the G4 count has been normalized for the genome size, as stated in the Methods.

3) However, the data have not been normalized for the GC content of the genome. The authors state that the GC content did not basically vary among species: but as shown in fig 2a and table S1, the difference gets up to more than double (19% to 46%, which is a lot). This may be a partial explanation why *Macrostomum lignano* (46% GC content) has G4 density higher than mammals, which the authors considered an exception (top of page 8). Therefore, the data and the G4 count have to be normalized for the GC content.

4) In this direction, it also necessary to count the number of G4s that would statistically form in a random sequence with the composition of each genome under study and compare it with the actual number that is predicted for that species. This will help assess that the G4-forming sequences are not present by mere chance. It will also tell if there is a lower representation of G4-forming sequences than expected based on the sequence composition.

5) In fig 2b: please mention not only the density of the peaks but also the intensity and explain the reported data in this light.

6) The immunofluorescence data on the 4 species considered are very interesting and they provide experimental substantiation to the bioinformatics findings. These experimental data need to be extended to more of the 37 considered species, ideally to all of them, to prove the experimental correspondence with the informatics data.

7) Fig 5a is really difficult to read. In Fig 5a, b and especially c I don't understand the rational of loop combination sequential presentation: e.g. why the loop combination 6:7:6 is shown before 1:6:1? Is that considered to be less complex? Why? It seems to me that the loop composition in fig 5c is adjusted for the loop abundance into the different species, and not for the increasing complexity of the loops as it would be expected reading the text.

Other comments

8) In the Introduction please cite that G4s have been found also in prokaryotes and viruses besides eukaryotes. Computational predictions in viruses have been provided and a correlation between the composition of G4s in viruses and humans is available (doi: 10.3390/molecules241019429; doi: 10.1371/journal.pcbi.1006675.; doi: 10.1093/nar/gky187.). General data on prokaryotics are also available (doi: 10.3389/fgene.2019.01002). These have to be added to give the reader an overview on how G4s are spread in the kingdoms of life and

beyond.

9) At page 4, the haploid genome is mentioned: this is either a mistake or it is not clear at all. What about the diploid genomes that were studied?

10) At page 8 "red jungle fowl" is mentioned: this term is not mentioned anywhere else in the text and should be amended or explained that elsewhere it is indicated as Ggal.

11) page 8 bottom: amend the sequence as follows: "We further analysed the regions in the human genome with the highest....."

12) First sentence in the introduction: "-quadruplex (G4) structures and i-motifs are DNA tetraplexes that typically form in guanine-rich regions of genomes". either remove i-motifs or add cytosine-rich region. As it is, the sentence is not correct.

13) first sentence page 3: substitute predication with prediction

Responses to the reviewers' comments and concerns

Reviewer #1 (Remarks to the Author):

In this manuscript, Wu et al perform a bioinformatic search for G4 motifs in 37 different species. This study suffers from the following shortcomings: These 37 species are claimed to be “critical evolutionary nodes in the tree of life”... but there is not a single prokaryote or plant in this list! Why choose *Plasmodium reichenowi* and not *Plasmodium falciparum*? In addition, what is the evidence that these species are actually “representative”? Later in the ms, the authors select a subset of these 37 species (for example 9 in figure 2a) again claiming these are “representative species”.

A: We have modified the relevant sentences. In this study, we mainly focused on the evolution of G4 motifs in the lower organisms and higher animals, i.e., the primitive eukaryotes to animalia, rather than the whole tree of life because that there are a few studies reporting the G4 motifs of plant species (Capre et al., 2010; Garg et al., 2016). These species are in the phylogenetic nodes of animal evolution. We modified the related description in the revised manuscript. As to the term “representative”, we meant to refer the representative evolutionary nodes but not the species. For each evolutionary node, we selected two or three species with relatively intact genome information.

Both *Plasmodium reichenowi* and *Plasmodium falciparum* have genomic information with similar genome size and GC content. As we mentioned above and modified in our revised manuscript, we focused on the representative evolutionary node rather than species. Under this consideration, either *P. reichenowi* or *P. falciparum* appears to be suitable for G4 analysis in this study.

Regarding to the term of “representative”, we have modified the description. We chose nine species each from the phylogenetic categories to better demonstrate the evolutionary pattern of G4 distribution at the chromosome level along the large-scale evolutionary process. The “representative” is a relative concept and may not be very accurate. We have changed “nine representative species” to “nine selected species each from representing phylogenetic categories ranging from the fungus to mammal.” in the revised version.

References

- Capra, J. A., Paeschke, K., Singh, M., and Zakian, V. A. (2010). G-quadruplex DNA sequences are evolutionarily conserved and associated with distinct genomic features in *Saccharomyces cerevisiae*. *PLoS Comput. Biol.* 6:e1000861.
- Garg, R., Aggarwal, J., and Thakkar, B. (2016). Genome-wide discovery of G-quadruplex forming sequences and their functional relevance in plants.

Sci. Rep. 6:28211.

Table S1 - minus the last column - could be moved to the main MS. In this Table, the link to *Octopus anatinus* is wrong (the one listed corresponds to *Octopus bimaculoides*). The low number of genes in this species is surprising. How reliable are the numbers of genes provided for each species? 42,454 in humans vs only 2,717 in *Octopus anatinus*?

A : Thanks for your suggestion. We have modified Table S1 with the statistic information of G4 motifs and generated a new table (Table 1) in the revised manuscript.

Sorry for our mistake. The species therein should be *Ornithorhynchus anatinus*, which belongs to mammal and has 33,610 genes. We are sorry for the typos and the wrong link. We have double checked and confirmed that the genome sequence and the annotation information. The typos and the link are corrected (http://asia.ensembl.org/Ornithorhynchus_anatinus/Info/Index).

Also annoying are statements implying that the proportion of G4 bearing genes increased “with species evolution”. A very simple organism can result from a highly sophisticated evolution process and should not be considered primitive.

A: Thanks for your comment. Surely, a very simple organism can result from a highly sophisticated evolution process in some cases. The evolutionary position of an organism is determined by multiple characteristics. Here in our study, we followed the commonly recognized evolutionary process of animals. We changed our statements to “during the large-scale species evolution.”

Fig 1a: where does this tree of life come from? Provide a reference for the “general order of divergence during evolution”. Is there a consensus on the “general order of divergence” proposed here?

A: The figure is a schematic diagram that highlights the phylogenetic relationship of the 14 groups from fungus to mammals. We modified the figure based on previous study (Casewell et al., 2013). The general large-scale evolutionary pattern is well recognized. We added a reference (Hinchliff, et al. 2015) in our revised manuscript.

No, we cannot confirm the genetic relationship at species level. Therefore, we removed the species clade in the new figure. We removed the numbers on the tree and the statement: “The number following each phylogenetic category indicates the general order of divergence during evolution.”

Reference:

Casewell NR, Wuster W, Vonk FJ, Harrison RA, Fry BG. 2013. Complex cocktails: the evolutionary novelty of venoms. *Trends Ecol Evol.* 28(4):219-29.

Hinchliff CE, Smith SA, Allman JF, Burleigh JG, Chaudhary R, Coghill LM, Crandall KA, Deng J, Drew BT, Gazis R, et al. 2015. Synthesis of phylogeny and taxonomy into a comprehensive tree of life. *Proc Natl Acad Sci U S A* 112:12764-12769.

The limitations of quadparser are not discussed – more accurate prediction tools are available. I strongly suggest the authors to read the guide to computational methods for G4 predictions recently published in NAR (doi: 10.1093/nar/gkz1097)

A: Thanks very much for your information. We have read that paper by Puig Lombardi and Londono-Vallejo, which was published this year and we did not read it when we prepared this manuscript. In addition, we also noted that recently, a lot of newly developed tools based on different algorithms to predict quadruplexes of substantially new types, which are beyond the prediction by the classical algorithm, i.e., Quadparser. However, the potential G4s predicted by those tools may be lack of experimental evidence, or the potential G4s predicted might be unstable, such as G4Hunter.

According to the suggestion from the reviewer and that study, we extended our analysis to the loop length (1-12) of the G4 motifs using Quadparser in the revised manuscript. Consistently, the similar results, i.e, the number, density, total length ratio and the proportion of genes that bearing G4 motifs in the upstream 2kb region were generally increased in the 37 species of the 14 phylogenetic categories during the large-scale evolution, as shown in the below and new Figure 1. The results and conclusions are not changed.

G4 motifs summary. (A) density, normalized density, length ratio of GC3L3112 motifs and the proportion of the genes bearing GC3L3112 motifs in the upstream 2 kb region in the genomes of 37 species of the 14 phylogenetic categories. (B) those of GC3L317 motifs. R2: goodness of fit of the trend lines. *, p<0.05; **, p<0.01; and ***, p<0.001 by F test.

With the increase of loop length, the stability of G4 motifs will decrease. Although there might be false negative, the canonical type of G4 motifs with loop length ranging from 1-7 is stable. We then used this type of G4 motifs as an example for further analyses.

The limitations of Quadparser probably is its highly stringency in prediction of G-quadruplexes, which would cause false negative. It may not efficiently detect some imperfect or non-canonical quadruplexes. Thus, it may not be sufficient for the preliminary and reliable screening of low-abundant potential conformations or sequences by computational analysis. However, in this study, we need reliable and conservative, not aggressive, prediction. In the

revised manuscript, we added a paragraph to discuss this:

“In this study, large-scale evolutionary patterns of the genomic G4 motifs were analysed for selected species of organisms from single-celled eukaryotic fungi to humans by using the classical algorithm quadruplexes (34). Recently, a lot of newly developed tools based on different algorithms have been used to predict new types of quadruplexes, which are beyond the prediction by Quadparser (36). The limitations of Quadparser probably is its highly stringency in prediction of G4, which may cause false negative. It may not efficiently detect some imperfect or non-canonical quadruplexes. Thus, it may not be sufficient for the preliminary and reliable screening of low-abundant potential conformations or sequences by computational analysis. In this study, because our core interest is the evolutionary pattern of G4s, the stable and credible type of G4 motifs, i.e, (G/C)3L1-7, was used to predict. The types of G4 motifs with longer loop length, such as 1-12, were also analysed. The results indicated that G4 motifs with longer loop length showed similar genomic distribution pattern with (G/C)3L1-7 type, (Fig. S3).”

“All four types of potential G4 motifs were included” but i) one could define other types (longer loop lengths have been tested; see reference cited in the previous and ii) the interest of each of these 4 types is not explained. When counting the number of G4 motifs, how do the authors treat overlapping motifs (for example does (GGGTTA)₅ count for one or two quadruplexes?).

A: As to G4 structure, the number of Gs in G-tract could be 2, 3, or 4 (Bugaut and Balasubramanian, 2008; Puig Lombardi and Londono-Vallejo, 2020). Because when there are two Gs in a G-tract, the more stable G4 structure tends to have the loop with length of 1-4bp (Bugaut and Balasubramanian, 2008), we included these types for analysis. As we mentioned above, we also added analysis of more types of G4 structure (4 and 5 Gs in a G-tract and extended the loop length to 12 bp) in the revised manuscript. The reason for analyzing these types of potential G4 motifs is that they almost reliably cover most G4 motifs.

We used the software Quadparser to identify G4 motifs, in which the overlapping motifs were merged as one.

We added the information to the revised manuscript.

Reference:

Bugaut, A., Balasubramanian, S., 2008. A sequence-independent study of the influence of short loop lengths on the stability and topology of intramolecular DNA G-quadruplexes. *Biochemistry* 47, 689-697.

Puig Lombardi, E. and A. Londono-Vallejo, 2020. A guide to computational methods for G-quadruplex prediction." *Nucleic Acids Res* 48(1): 1-15.

Genome annotation reliability (for gene definition, transcription factors) and even assembly (because of repetitive motifs) may differ between species – the authors never discuss this point.

A: Genome annotation reliability are largely affected by the quality of genome assembly. In our study, we tried to select those species with high-quality sequenced genomes. This is also one of the reasons why we used one species but

not the other species in a phylogenetic category. We cannot guarantee all the genome sequences reported by other laboratories are perfectly or totally accurate, but we believe that the overall trend of the analysis results is reliable.

In this study, all the genomic sequence data are from related databases as we described in Materials and Methods and gene definition for transcription factors is based on the resources in the public Gene Ontology database (<http://geneontology.org/>). Gene annotation and definition have recognized standard in the scientific community (Yates et al., 2020). Actually, we could not exclude the any possible mistakes in genome assembly and gene annotation by other groups.

We have added above discussion in the revised manuscript.

Reference:

Yates AD et al. *Ensembl* 2020. *Nucleic Acids Res.* 48(D1):D682-D688

Fig 2a: It would be more informative to plot the density of G4 motifs vs GC content for all species.

A: Thanks for your suggestion. Yes, it would be better to consider GC content when G4 density is analyzed. According to the editor and the reviewers' comments, we have normalized G4 density over the GC content (Fig 2a) for all species (see Materials and Methods and Fig 2a in the revised manuscript). In addition, we updated Fig 2d by adding the window plotting information of CG content along the genomes of the nine species (Fig 2d). The results indicate that when normalized with GC contents, the density of G4 motifs also showed a general increase along the species evolution. Thus, the effect of the genomic GC content is not the major factor contributing to the observed increase in G4 density in the genomes during species evolution.

Fig 2b is useless at this magnification. What is the central circling arrow supposed to mean? Evolution time??

A: This figure shows the clusters and the cluster densities of the G4 structures in the chromosomes of these species. The radial lines in the circles show the cluster density and intensity in each analysis windows. We have re-done Figure 2 and used the high-resolution version of Fig 2d. We removed the central circling arrow, which originally meant the direction of evolution, and added GC content and density in the revised figure.

Fig 3: was a trivial control (adding DNase) performed to make sure these foci correspond to G4 DNA ? Fig 3a is a rather trivial control that only needs to be showed in sup info. The problem to count BG4 foci number is that the fluorescent spots have different intensities. Was a threshold level defined?

A: We have added a control in that the cells were first treated with DNase I. The results showed that DNase I treatment significantly decreased the number of the staining foci in the nuclei (Figure 3). This result indicates that these foci are G4 DNA.

The original Figure 3a has been moved to Supplementary Information (Figure S2).

The method of counting foci refers to Zeraati's paper (Zeraati et al., 2018). Yes, it is difficult to have a definite and absolute threshold to count a focus. It is somehow empirical. However, the key here is that the chosen threshold was always kept the same and consistent in all the samples of all the experiments so that the comparison can be conducted. In this study, we counted the spots in the same magnification under a fluorescence microscopy for all the samples.

Reference:

Zeraati, M., Langley, D.B., Schofield, P., Moye, A.L., Rouet, R., Hughes, W.E., Bryan, T.M., Dinger, M.E., Christ, D. (2018) I-motif DNA structures are formed in the nuclei of human cells. *Nat. Chem.*, 10, 631-637.

Fig 6: Are pig and silkworm methylations pooled together? What is the fraction of G4 genes in both cases?

A: No. We analyzed pig and silkworm methylation separately. a, b are the results from pig and c, d are the results from the silkworm. The fraction of G4 genes in pig is 0.3656 and that in silkworm is 0.0227.

Fig 6c is misleading (and histogram starts at 0.95!). It would be more informative to plot the % of mC in both cases.

A. Sorry for the misleading. We modified Fig 6c to make it clear. In the case of Fig 6c, the significant difference comes from the proportion of Un_mC and mC in all genes (All) and genes bearing GC3L317 motif. We showed both the Un_mC and mC in the revised manuscript.

Table 2 is of little interest in the main MS – should be move to S.I.

A: We have moved it to S.I (Table S3) in the revised manuscript.

Reviewer #2 (Remarks to the Author):

In this manuscript Wu et al study the presence of G-quadruplex structures (G4s) in 37 different species representative of the different evolutionary stages of the eukaryotic genome. By a bioinformatics approach they show that the abundance and complexity of G4s augmented with the evolution and with the increasing complexity of the considered eukaryotic organisms. They show the abundance of transcription factors among genes bearing G4s and that methylation sites are antagonized by the presence of G4-forming sequences. By immunofluorescence they visualized the abundance of G4 in 4 representative species.

Overall the findings are of general interest and further support the G4-mediated regulation of cellular genes in organisms. The manuscript is rationally organized and results clearly presented.

A: Thanks very much for your positive comments.

I would suggest addressing the following major and minor comments to improve the manuscript.

Major comments

1) The authors state: “All four types of potential G4 motifs were included: (G/C)2L1-2, (G/C)2L1-4, (G/C)3L1-3 and (G/C)3L1-7 “. However, there are many more potential types, with longer loops (as the authors also show) and with a higher number of G/C layers (e.g. 4). Please revise this throughout the text.

A: Thanks for your suggestion. We have extended loop range (i.e, 1-12 bp) and added higher number of G/C layers (4 and 5) in our analysis in the revised manuscript. The sentence have been changed to “Therefore, the types of potential G4 motifs were included: (G/C)2L1-4, (G/C)3L1-7, (G/C)3L1-12, (G/C)4L1-12 and (G/C)5L1-12...” The new data have been presented in the text and figures.

2) At page 7 the authors state: “...suggesting that the increase in the number of G4s in the genomes was not only because of the increase in genome size but also...”. This is not correct because the increase in the genome size cannot be considered since the G4 count has been normalized for the genome size, as stated in the Methods.

A: Thanks for your comments. What we wanted to say is that the absolute number of G4 increased with the genome size, but the density is not positively related to the genome size. Increase in the genome size provides more G and C bases, therefore more chance for the formation of the G4 structures. The sentence has been modified: “suggesting that although the increase in the number of G4s in the genomes was partially associated to the increase in genome size, the increase in the density of G4s was due to the increase in species complexity”

Based on the editor and other reviewer’s suggestion, we generated a simulation test. Consequently, we found that increase in the G4 density is not because of randomly increased genome size (Figure S1). It is quite possible to be a biological feature in the large-scale genomic evolution.

3) However, the data have not been normalized for the GC content of the genome. The authors state that the GC content did not basically vary among species: but as shown in fig 2a and table S1, the difference gets up to more than double (19% to 46%, which is a lot). This may be a partial explanation why *Macrostomum lignano* (46% GC content) has G4 density higher than mammals, which the authors considered an exception (top of page 8). Therefore, the data and the G4 count have to be normalized for the GC content.

A: Thanks for your suggestion. Based on your suggestion, we normalized G4 density over the GC contents (%) in each species (Figure 2b, c) in the revised manuscript. We found that the general increased patterns of G4 density are similar to the original patterns. We added these data (Figure 2b, c) and information in our revised manuscript.

As to why *Macrostomum lignano* (46% GC content) has G4 density higher than mammals, GC content may have an influence. However, when its G4 density was normalized over the GC content, we found that it is lower than that of mammals.

4) In this direction, it also necessary to count the number of G4s that would statistically form in a random sequence with the composition of each genome under study and compare it with the actual number that is predicted for that species. This will help assess that the G4-forming sequences are not present by mere chance. It will also tell if there is a lower representation of G4-forming sequences than expected based on the sequence composition.

A: Thanks for your suggestion. We have generated the simulation with an in-house perl script. Briefly, for each genome, we generated a simulated genome with the same genome size to the related real genome while the four bases (A, T, C or G) are randomly used at each site. We found that in the randomly simulated genomes, G4 counts generally increased along with the large-scale evolution, similar to the pattern of the real genomes (see the figure below, left panel). However, when comparing the G4 density, the simulated genomes showed drastic different patterns from the real ones. The G4 density is generally similar in among the simulated genome, but the G4 density in the real genomes show general increment along with the large-scale evolution (see the figure below, right panel). We added related results and discussion in the revised manuscript.

5) In fig 2b: please mention not only the density of the peaks but also the intensity and explain the reported data in this light.

A: Thanks for your suggestion. Yes, we have mentioned not only the density but also the intensity of the peaks was increased with species evolution in the revision.

We plotted the density of G4 motifs along the genome with 50kb sliding window. From the plotting, we could find that in the two species with high G4 density, i.e., the Ggal (Red jungle fowl) and Hsap (human), the G4s are not evenly distributed. The high-density peaks are somewhat mosaic in along the chromosomes. As to those with quite low G4 density, the distribution is general even.

6) The immunofluorescence data on the 4 species considered are very interesting and they provide experimental substantiation to the bioinformatics findings. These experimental data need to be extended to more of the 37 considered species, ideally to

all of them, to prove the experimental correspondence with the informatics data.

A: Thanks for your suggestion. Most of the 37 species are difficult to obtain or technically difficult in immunofluorescence experiment. We added another three species (*Pelodiscus sinensis*, *Gallus gallus*, *Ovis aries*) in immunofluorescence experiment (Figure 3d-f). As the first version of the manuscript, these additional results showed that the number of G4 structures in the nuclei increased during the large-scale species evolution.

7) Fig 5a is really difficult to read. In Fig 5a, b and especially c I don't understand the rationale of loop combination sequential presentation: e.g. why the loop combination 6:7:6 is shown before 1:6:1? Is that considered to be less complex? Why? It seems to me that the loop composition in fig 5c is adjusted for the loop abundance into the different species, and not for the increasing complexity of the loops as it would be expected reading the text.

A: Sorry for the vague description. The software quadparser has the function to calculate the length of each loop in the G4 motif. For example, 6:7:6 indicates the length of the first: second: third loop, respectively. The base numbers were segregated by ":". The order of the loop length composition is based on the number of species that have certain type of loop length composition. The Y-axis shows the numbers of the species that have a specific loop length composition in the X-axis. For example, there are three species, out of 37 species, containing 6:7:6 loop length type of G4, while there are two species containing 1:6:1 type of G4. This figure also shows that most species, either lower or higher organisms, have G4 structures with short loops, while the G4 structures with longer loops were probably emerged with the species evolution.

Other comments

8) In the Introduction please cite that G4s have been found also in prokaryotes and viruses besides eukaryotes. Computational predictions in viruses have been provided and a correlation between the composition of G4s in viruses and humans is available (doi: 10.3390/molecules241019429; doi: 10.1371/journal.pcbi.1006675.; doi: 10.1093/nar/gky187.). General data on prokaryotes are also available (doi: 10.3389/fgene.2019.01002). These have to be added to give the reader an overview on how G4s are spread in the kingdoms of life and beyond.

A: We have added the references in Introduction of the revised manuscript. Here in this study we want to address the evolutionary pattern of G4s from primitive eukaryotes to Animalia, rather than all evolutionary nodes in the tree of life indeed. Yes, it would be helpful to let the readers know the works in prokaryotes and viruses.

9) At page 4, the haploid genome is mentioned: this is either a mistake or it is not clear at all. What about the diploid genomes that were studied?

A: Thanks. It was a mistake and the word "haploid" has been deleted.

10) At page 8 "red jungle fowl" is mentioned: this term is not mentioned anywhere

else in the text and should be amended or explained that elsewhere it is indicated as Ggal.

A: Thanks for your suggestion. Red jungle fowl is the common name for Ggal. We have explained it at its first appearance.

11) page 8 bottom: amend the sequence as follows: “We further analysed the regions in the human genome with the highest.....”

A: Thanks. We have amended it.

12) First sentence in the introduction: “-quadruplex (G4) structures and i-motifs are DNA tetraplexes that typically form in guanine-rich regions of genomes”. either remove i-motifs or add cytosine-rich region. As it is, the sentence is not correct.

A: Thanks. We have removed i-motifs.

13) first sentence page 3: substitute predication with prediction

A: Thanks. We have substituted it.

REVIEWERS' COMMENTS:

Reviewer #1 (Remarks to the Author):

The authors have made laudable efforts to address my criticisms. While I still do not agree with some of the answers and conclusions, this work may be of interest for researchers working on quadruplexes in non human species.

Minor point: it seems one question remained unanswered:

"When counting the number of G4 motifs, how do the authors treat overlapping motifs (for example does (GGGTTA)₅ count for one or two quadruplexes?)"

Reviewer #2 (Remarks to the Author):

The authors have satisfactorily answered to my previous comments.

REVIEWERS' COMMENTS:

Reviewer #1 (Remarks to the Author):

The authors have made laudable efforts to address my criticisms. While I still do not agree with some of the answers and conclusions, this work may be of interest for researchers working on quadruplexes in non human species.

A: Thanks very much for your positive comments.

Minor point: it seems one question remained unanswered:

"When counting the number of G4 motifs, how do the authors treat overlapping motifs (for example does (GGGTTA)₅ count for one or two quadruplexes?)"

A: The software quadparser (version 2.0) merges the overlapping quadruplex motifs as one. We test the sequence (GGGTTA)₅ by quadparser 2.0. Consistently, only one G4 motif was predicted: GGGTTAGGGTTAGGGTTAGGG. Meantime, the software also indicates two overlappings.

Reviewer #2 (Remarks to the Author):

The authors have satisfactorily answered to my previous comments.

A: Thanks very much for your positive comments.